# Learning to Describe Urban Change: Graph-Guided Detection and spatio-Temporal State Space Model with Uncertainty Estimation

## Abstract

Automated change detection (CD) and captioning from satellite imagery plays a crucial role in urban development monitoring, infrastructure assessment, and land-use analysis. However, existing change captioning systems lack uncertainty quantification, making it challenging to assess prediction reliability when analyzing critical infrastructure changes, building construction, or environmental modifications where inaccurate interpretations could impact urban planning decisions or infrastructure management. We address this limitation through a comprehensive pipeline combining SemanticGraphCD module for enhanced change detection with a State Space Model(SSM)-based captioning module for scalable description generation. SemanticGraphCD integrates graph neural networks with task-agnostic semantic learning, employing an adaptive processing mechanism that dynamically switches between GNN-based feature propagation and convolutional operations. This architecture learns semantic representations through bi-temporal consistency constraints, better discriminating meaningful infrastructure and land-use changes from temporal variations in very high-resolution imagery. The State Space Model based captioning module contains a Spatial Difference-aware SSM (SD-SSM) which improves upon previous CNN and Transformer-based models in receptive field. Moreover a Temporal Traversing SSM (TT-SSM) is used which scans bi-temporal features in a temporal cross-wise manner enhancing the model's temporal understanding and information interaction. This SSM is guided by SemanticGraphCD's change masks using a convolutional focusing module which aggregates change information from the masks with the bitemporal images. This guides the model in representing the changes between the bi-temporal images within the state space model hidden states, enabling linear computational scaling while maintaining competitive performance. Instead of treating all caption tokens equally in the context of remote sensing, we introduce Semantic-Weighted Sentence Entropy (SWSE) for principled uncertainty quantification. SWSE emphasizes domain-relevant vocabulary over function words, providing interpretable confidence measures that correlate with caption quality. Experimental results demonstrate that our approach achieves improvement in captioning performance compared to existing state space models, while SWSE provides reliable uncertainty estimates for informed decision-making in urban monitoring applications.

## 1 Introduction

The automated analysis of satellite imagery forms the backbone for global monitoring efforts, supporting applications in disaster response, urban planning, infrastructure assessment, and environmental management. With the increasing availability of high-resolution satellite data from missions like Landsat Wulder et al. (2019), Sentinel Drusch et al. (2012), and commercial providers Li et al. (2022b), there is unprecedented opportunity for continuous Earth observation. Within this domain, change detection (CD) has emerged as a key technique, identifying differences between bi-temporal images to reveal events such as building construction, deforestation, or road expansion Demir et al. (2013); Ertürk et al. (2017); De Alban et al. (2018). Beyond merely detecting change, the task of

change captioning seeks to generate natural language descriptions that summarize the most meaningful differences, enabling human-interpretable insights for decision-makers Hoxha et al. (2018); Shi et al. (2022). The evolution of change detection methods has progressed from traditional pixel-based approaches Radke et al. (2005) to sophisticated deep learning architectures. Early methods relied on simple differencing or thresholding techniques Singh (1989), which were limited by their sensitivity to noise and inability to capture semantic changes. The introduction of object-based change detection Blaschke et al. (2008) and machine learning approaches Lu et al. (2004) improved robustness , but still required manual feature engineering. Deep learning revolutionized the field with convolutional neural networks (CNNs) Zhang & Li (2017); Daudt et al. (2018) that could automatically learn hierarchical features, followed by more advanced architectures like U-Net variants Peng et al. (2019) and attention mechanisms Chen & Shi (2020).

Existing change captioning approaches have made progress by combining deep change detection modules with natural language generation architectures. Attention-based methods, including Siamese neural networks Chang & Ghamisi (2023a) and Sparse Focus Transformers (SFTs) Sun et al. (2024), improve the localization of changes by focusing on the most relevant regions, but often at high computational cost or at the risk of missing small, distributed changes that require dense modeling. Vision-language models have shown promise in general image captioning Xu et al. (2015); Anderson et al. (2018), leading to adaptations for remote sensing applications Lu et al. (2018); Ramos et al. (2023). More recently, state space models such as Mamba Gu & Dao (2023) have demonstrated efficiency in modeling long-range spatio-temporal dependencies Qi et al. (2023), while change-guided approaches Zheng et al. (2022) leverage binary masks to explicitly highlight regions of change before caption generation. Graph neural networks have gained attention for their ability to model spatial relationships in remote sensing data Hong et al. (2021); Wan et al. (2019). Several works have explored GNNs for change detection Song et al. (2022); Tang et al. (2022), demonstrating their effectiveness in capturing contextual information and spatial dependencies. However, the computational complexity of GNNs on dense imagery remains a challenge, motivating hybrid approaches that balance accuracy with efficiency Liu et al. (2022b).

The challenge of uncertainty quantification in machine learning has received significant attention across various domains Gal & Ghahramani (2016); Lakshminarayanan et al. (2017). In computer vision, uncertainty estimation has been explored for object detection Laplace et al. (2021), semantic segmentation Kendall & Gal (2017), and image classification Sensoy et al. (2018). For natural language generation, uncertainty quantification has been studied in machine translation Wang et al. (2019) and text summarization Zhang et al. (2020), but remains underexplored in vision-language tasks, particularly in remote sensing applications where reliability is crucial for decision-making Robinson et al. (2017).

However, most existing systems neglect an equally important aspect: uncertainty quantification. In safety-critical applications like infrastructure monitoring, disaster response planning, and urban development assessment, unreliable or overconfident captions can lead to poor planning decisions, misallocation of resources, or inadequate emergency responses Voigt et al. (2016); Plank (2014). The high-stakes nature of these applications demands not only accurate predictions but also reliable confidence estimates that enable human experts to assess when model outputs should be trusted Jiang et al. (2018).

In this work, we address these challenges with a unified pipeline that couples a SemanticGraphCD module for robust change representation learning with a State Space Model (SSM)-based captioning module for scalable description generation. SemanticGraphCD integrates graph neural networks with task-agnostic semantic learning through an adaptive processing mechanism that dynamically switches between GNN-based feature propagation and convolutional operations. This architecture learns semantic representations via bi-temporal consistency constraints to better discriminate meaningful infrastructure and land-use changes from temporal variations. Our SSM-based captioning module incorporates Spatial Difference-aware SSM (SD-SSM) and Temporal Traversing SSM (TT-SSM) components that enhance temporal understanding while enabling linear computational scaling, addressing the quadratic complexity limitations of transformer-based approaches Vaswani et al. (2017).

Critically, we introduce Semantic-Weighted Sentence Entropy (SWSE), a principled sentence-level uncertainty measure that assigns greater importance to domain-relevant content words over function words, yielding interpretable confidence scores aligned with caption quality. Unlike existing

uncertainty measures that treat all tokens equally Malinin & Gales (2018) , SWSE recognizes that uncertainty in semantically important terms (e.g., "building", "residential") is more concerning than uncertainty in function words (e.g., "the", "has"). Together, these contributions provide more accurate and reliable captions with trustworthy uncertainty estimates for urban monitoring and decision-support systems. In summary, our main contributions are as follows:

1. We propose a novel change detection backbone that combines graph neural networks with convolutional operations via an adaptive processing mechanism. This hybrid approach captures long-range spatial dependencies while remaining computationally tractable. Bi-temporal consistency constraints are used to learn semantically meaningful representations that better distinguish infrastructure and land-use changes from irrelevant temporal variations.

2. We adopt a State Space Model (SSM)-based captioning module incorporating two key components: (a) a *Spatial Difference-aware SSM (SD-SSM)*, which enlarges the effective receptive field and improves spatial sensitivity to subtle changes, and (b) a *Temporal Traversing SSM (TT-SSM)*, which scans bi-temporal features cross-wise, enhancing temporal understanding and information interaction. Together, these modules achieve linear computational complexity while outperforming transformer-based approaches on change captioning tasks.

3. We introduce a convolutional focusing module that leverages change masks from SemanticGraphCD to guide the SSM hidden states. This explicitly emphasizes regions of interest, improving the alignment between visual changes and their corresponding textual descriptions.

4. We propose a novel sentence-level uncertainty metric that assigns higher weights to domain-relevant content words (e.g., *building*, *road*) while down-weighting function words. This yields interpretable and task-aware confidence scores that correlate with caption quality, providing actionable reliability estimates for decision-making in urban monitoring applications.

## 2 METHODOLOGY

We have implemented a three-stage architecture consisting of (i) a change detection module using SemanticGraphCD with graph neural networks and task-agnostic feature learning to generate semantic change masks, (ii) a change extraction module with image enhancement (IE Module), CLIP ViT-B/32 backbone Radford et al. (2021), and dual state space models (SD-SSM and TT-SSM) for spatio-temporal modeling, and (iii) a language decoder for caption generation with integrated Semantic Weighted Sentence Entropy (SWSE) for enhanced interpretability Figure 1.

### 2.1 CHANGE DETECTION MODULE

Our change detection module employs SemanticGraphCD which incorporates adaptive processing that dynamically switches between graph neural network-based feature propagation and convolutional operations. Given bi-temporal remote sensing images, the module extracts multi-scale features through a CNN backbone, processes them through both graph networks for semantic relationships and task-agnostic feature learning components, then uses an attention fusion mechanism and change detection head to generate binary change masks. This approach effectively discriminates meaningful infrastructure and land-use changes from temporal variations by learning semantic representations through bi-temporal consistency constraints.

### 2.2 CHANGE EXTRACTION MODULE

The change extraction module processes bi-temporal images and generates change masks through three sequential components following the architecture shown in Figure 1.

**Image Enhancement (IE Module).** We implement mask-guided image fusion where binary change masks undergo element-wise multiplication with the original bi-temporal images. To address blank

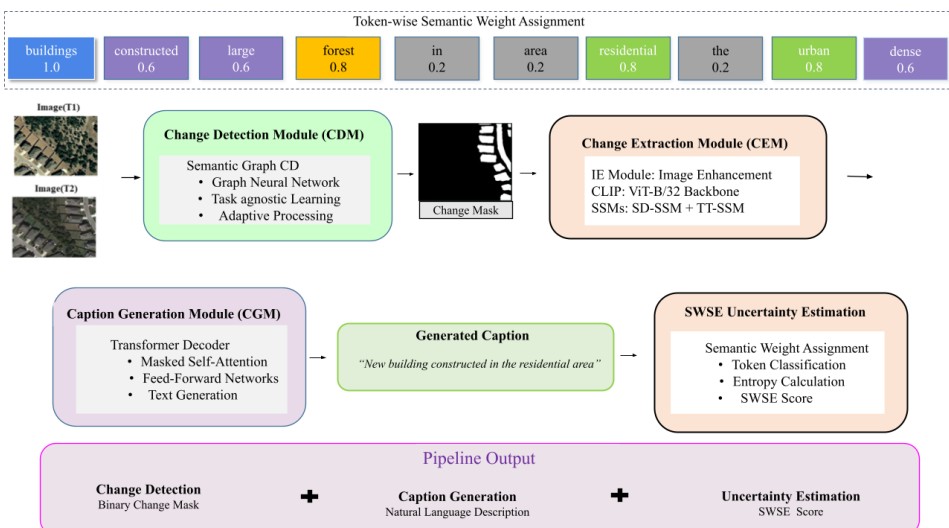

Figure 1: **Overview of the proposed three-stage architecture for change detection and captioning.** The pipeline consists of: (1) CDM that uses SemanticGraphCD with graph neural networks to generate change masks, (2) CEM that enhances images and processes features via dual state space models, and (3) CGM that generates natural language descriptions. Outputs include change masks, captions, and SWSE confidence scores

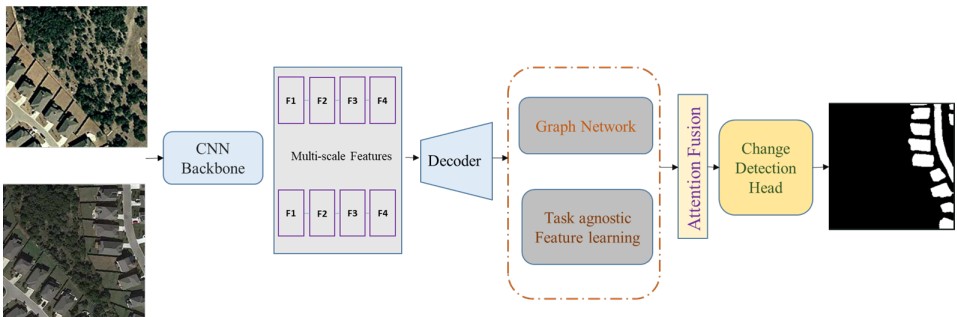

Figure 2: **SemanticGraphCD architecture for change detection.** The framework extracts multi-scale features (F1-F4) via CNN backbone, processes them through parallel Graph Network and Task-agnostic Learning modules, then uses attention fusion and Change Detection Head to generate binary change masks.

mask issues common in challenging samples, the IE module includes an adaptive fallback that returns original images when masks contain insufficient change information. This enhancement provides better spatial information regarding changed objects, guiding the model to improve accuracy when describing changes.

**CLIP Backbone.** We utilize the frozen CLIP ViT-B/32 image encoder to extract robust visual representations from the enhanced bi-temporal images. The choice of CLIP over domain-specific encoders follows recent success in remote sensing applications and provides strong transferability across diverse geographical regions. We choose the image encoder over video encoders for two reasons: (1) Prior change detection approaches have demonstrated that Siamese encoders with weights shared across time are highly effective for identifying changes in sequences of Earth observation data Li et al. (2021b), and (2) image encoders provide more flexibility for variable sequence lengths while maintaining computational efficiency.

**State Space Models.** The extracted features are processed through dual state space models: Spatial-Difference SSM (SD-SSM) and Temporal-Transition SSM (TT-SSM) for joint spatio-temporal modeling. This design choice addresses the quadratic complexity limitations of traditional attention mechanisms when processing high-resolution remote sensing imagery, enabling efficient long-range dependency modeling with linear complexity. The SD-SSM captures spatial relationships between change regions, while TT-SSM models temporal transitions between bi-temporal features.

## 2.3 LANGUAGE DECODER

The language decoder follows a standard transformer architecture that inputs the spatio-temporal representations from the SSMs to generate natural language descriptions of detected changes. The decoder uses masked self-attention and feed-forward networks with residual connections for stable training.

## 2.4 SEMANTIC WEIGHTED SENTENCE ENTROPY (SWSE)

To enhance model interpretability and provide uncertainty quantification tailored to remote sensing applications, we introduce SWSE equation 1. Unlike classical Shannon entropy that treats all vocabulary tokens equally, SWSE assigns semantic importance weights based on domain relevance:

$$H_{SWSE}(X) = -\sum_{i=1}^{|V|} w_i \cdot p(x_i) \log p(x_i) \tag{1}$$

where weights $w_i \in \{0.2, 0.6, 0.8, 0.9, 1.0\}$ correspond to function words, descriptors, natural features, land use, and infrastructure categories respectively. This weighting ensures uncertainty over critical domain-specific terms contributes more significantly than uncertainty over common function words, providing meaningful confidence estimates for practical applications.

## 2.5 TRAINING DETAILS

We have used an Adam optimizer Kingma & Ba (2014) to train on NVIDIA RTX A5000 GPU. With an initial learning rate of 0.0001 and a step learning rate decay of 0.5 every 5 epochs. We used a batch size of 64, with the dimesion of word vectors being set to 768, and a beam size of 1. The number of multi-head attention mechanism is set to 8 and the model is trained for 50 epochs with validation done after every epoch and the model with the best performing BLEU-1 value getting its parameters saved. To evaluate model performance we use the following three metrics, BLEU-N(1,2,3,4) Papineni et al. (2002), CIDEr-D Vedantam et al. (2015) and ROUGE-L Lin (2004). BLEU-N measures how well a generated sentence matches the target sentence using n-grams precision, CIDEr-D (Consensus-based Image Description Evaluation, with damping) measures how consensus between candidate and multiple references. ROUGE (Recall-Oriented Understudy for Gisting Evaluation) measures how many n-grams or subsequences from the reference text appear in the generated text.

| Category | Weight ($w_i$) | Description | Example Tokens |
|---|---|---|---|
| **Infrastructure** | 1.0 | Core man-made structures whose changes are central to remote sensing analysis. | building, road, airport, bridge, dam, port, railway |
| **Land Use** | 0.9 | Human activity categories that reflect economic, social, or developmental shifts. | residential, commercial, agricultural, industrial, barren |
| **Natural Features** | 0.8 | Environmental elements that set the scene and often change alongside human impact. | forest, water, mountain, vegetation, river, coastline, glacier |
| **Descriptors** | 0.6 | Modifiers that qualify objects by size, condition, or density, adding nuance but not defining subjects. | many, large, dense, scattered, new, damaged, cleared |
| **Function Words** | 0.2 | Structural words essential for grammar but carrying little visual-semantic meaning. | the, this, has, and, in, a, is, of, with, from |

Table 1: **Semantic weight assignments in SWSE**
Higher weights are given to content-rich terms, while structural words receive lower values.

## 3 RESULTS

This section gives an evaluation of our proposed approach through extensive experiments on the LEVIR-CC and LEVIR-MCI datasets. We conduct quantitative comparisons with state-of-the-art change captioning methods, analyze the effectiveness of our SemanticGraphCD module, and demonstrate the utility of our proposed SWSE uncertainty metric for real-world remote sensing applications.

### 3.1 DATASET

In this work, we employ the LEVIR-CC dataset Liu et al. (2022a), the largest publicly available benchmark for remote sensing change captioning. The dataset consists of 10,077 pairs of $256 \times 256$ pixel images, comprising 5,039 unchanged pairs and 5,038 changed pairs, with temporal intervals ranging from five to fifteen years. Each image pair is described using five descriptive captions, where the captions for changed pairs are typically longer and more detailed than those for unchanged pairs. The standard split includes 6,815 pairs for training, 1,333 for validation, and 1,929 for testing. The vocabulary, derived from the training annotations, contains 463 unique words that appear more than five times and is augmented with four special tokens: *unk*, *start*, *end*, and *pad*.

The LEVIR-MCI dataset Liu et al. (2024), an extension of LEVIR-CC, provides pairwise temporal images together with multi-label change detection masks and descriptive sentences. It comprises 13,077 image pairs with corresponding multi-label masks. For the purpose of change captioning, the multi-label masks are converted into binary masks, denoting unchanged and changed pixels as 0 and 1, respectively. Each pair is annotated with five descriptive sentences, with explicit labels for roads and buildings.

We have also classified the entire vocabulary of LEVIR-CC as belonging to one of the 5 classes as described in Table 1. Figure 3 presents examples of bi-temporal images, their associated change maps, and captions generated by the model, where each word is color-coded according to its semantic category.

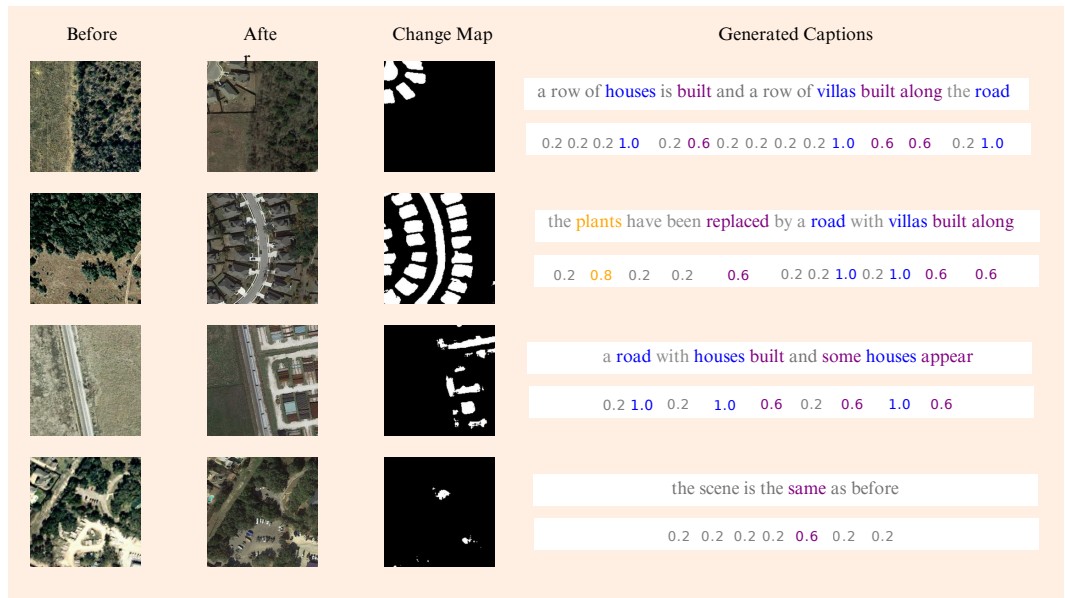

Figure 3: **Captions with semantic weighting.** Bi-temporal images with their corresponding change maps and generated change captions. Each caption is color-coded according to its semantic weighting category, with the associated weights shown below in the same colors. Blue denotes infrastructure, green denotes land use, orange denotes natural features, purple denotes descriptors, and gray denotes function words.

## 3.2 QUALITATIVE ANALYSIS

To verify the effectiveness of our model, we have compared results with various other state of the art change captioning models as shown in Table 2, i.e Capt-Rep-Diff Li et al. (2021a), Capt-Att Li et al. (2020), Capt-Dual-Att Li et al. (2022a), MCCFormer-S Li et al. (2023c), MCCFormer-D Li et al. (2023b), DUDA Li et al. (2023a) and PSNet Li et al. (2024). The Capt-Rep-Diff model uses a vision

| Method | BLEU-1 | BLEU-2 | BLEU-3 | BLEU-4 | ROUGE$_L$ | CIDEr-D |
|---|---|---|---|---|---|---|
| Capt-Rep-Diff | 72.90 | 61.98 | 53.62 | 47.41 | 65.64 | 110.57 |
| Capt-Att | 77.64 | 67.40 | 59.24 | 53.15 | 69.73 | 121.22 |
| Capt-Dual-Att | 79.51 | 70.57 | 63.23 | 57.46 | 70.69 | 124.42 |
| MCCFormer-S | 79.90 | 70.26 | 62.68 | 56.68 | 69.46 | 120.39 |
| MCCFormer-D | 80.42 | 70.87 | 62.86 | 56.38 | 70.32 | 124.44 |
| DUDA | 81.44 | 72.22 | 64.24 | 57.79 | 71.04 | 124.32 |
| PSNet | 83.86 | 75.13 | 67.89 | **62.11** | **73.60** | 132.62 |
| Ours | **83.93** | **73.21** | **68.01** | 60.32 | 73.01 | **133.23** |

Table 2: Quantitative evaluation of the proposed model in comparison with state-of-the-art approaches on the LEVIR-CC dataset. Results are reported across BLEU-1 to BLEU-4, ROUGE$_L$, and CIDEr-D metrics, with the best scores highlighted in bold.

transformer to extract features by employing progressive difference perception layers to obtain multiscale visual features. These features are then aggregated by a scale-aware reinforcement learning module and a transformer decoder to generate a textual description. The Capt-Att model utilizes a visual attention mechanism to focus on salient regions of the image, extracting key features that are then passed to a transformer-based decoder to generate the final description. The Capt-Dual-Att model extends this by incorporating a dual-attention mechanism, using both visual and semantic attention to better align the extracted features with the generated text. The MCCFormer-S model is a multi-modal cross-attention transformer that uses a single-stream approach to fuse image and text features for description generation. The MCCFormer-D model builds on this by employing a

dual-stream architecture, processing visual and textual information in parallel before fusing them with a cross-attention mechanism. The DUDA model, or "Dual-stream Unifying Dialogue-based Attention," uses a unique dual-stream architecture with an attention mechanism designed to unify information from both image and text streams. The PSNet model, or "Prompt-based Sentence Generation Network," uses a vision transformer to extract features by employing progressive difference perception layers to obtain multiscale visual features. These features are aggregated by a scale-aware reinforcement learning module and transformer decoder to generate a textual description.

The Table 2 demonstrates the performance of each model compared to our model. To highlight the best performance of each model, we have taken the experimental results of these models directly from their papers. These results indicate that compared to the previously mentioned methods, our model achieves superior performance overall, including BLEU-1,2 and 3 as well as CIDEr-D of 83.93%, 73.21%, 68.01%, and 133.23% respectively.

| Model | Change acc | No-change acc | Total acc |
|---|---|---|---|
| Chg2Cap | 88.28% | 97.72% | 93.00% |
| SEN | 85.06% | 97.82% | 91.44% |
| SparseFocus | 87.86% | **98.03**% | 92.95% |
| RSICCFormer | **90.91**% | 94.48% | 92.70% |
| Our Model | 90.21% | 96.04% | **93.13**% |

Table 3: **Comparison of change detection models on the LEVIR-CC dataset.** Results are reported for change accuracy, no-change accuracy, and overall accuracy. Our model achieves the highest overall accuracy.

Table 3 presents the results of our model and that of Chang & Ghamisi (2023b), SEN Wang et al. (2018), SparseFocus Zhai et al. (2025) and RSICCFormer Lu et al. (2023) for the task of determining whether changes exist in the bi-temporal image pairs, we can see that most models are good at either detection of image pairs with changes or those with no chnages, our model is more balanced with a slight preference for images with no change, it also outperforms all other models in overall accuracy.

To compare the impact of using ground truth masks vs masks generated by Semantic graph CD, analysis has been carried out by training using binary masks provided by LEVIR MCI as shown in Table 4. The results demonstrate that manually annotated masks, while inherently more accurate and serving as an upper bound, yield higher performance compared to automatically generated ones. Nevertheless, masks produced by Semantic Graph CD offer a scalable and annotation-free alternative, making the approach more practical for large-scale applications.

| Mask Source | BLEU-1 | BLEU-2 | BLEU-3 | BLEU-4 | ROUGE-L | CIDEr-D |
|---|---|---|---|---|---|---|
| Ground Truth (LEVIR-MCI) | 85.82 | 77.65 | 70.26 | 64.32 | 73.56 | 136.03 |
| SemanticGraphCD | 83.93 | 73.21 | 68.01 | 60.32 | 73.01 | 133.23 |

Table 4: **Impact of change mask quality on captioning performance.** Ground truth vs. generated masks

Table 4 presents a performance comparison between ground truth masks from LEVIR-MCI and masks generated by our SemanticGraphCD module. Ground truth masks achieve slightly higher scores, with a 4.0 BLEU-4 advantage, reflecting the benefit of manual precision. However, SemanticGraphCD delivers competitive results across all metrics, demonstrating its ability to generate reliable change cues without manual supervision. This validates our integrated pipeline as a practical alternative that balances accuracy with scalability. This makes our approach especially suitable for large-scale remote sensing applications where manual mask creation is impractical.

Table 5 compares uncertainty quantification between standard entropy and our proposed SWSE metric. RSICCFormer exhibits the lowest standard entropy (0.53), indicating high model confidence, yet maintains comparable SWSE values (0.56), suggesting that its uncertainty is appropriately concentrated on semantically important terms. RSCaMa shows moderate standard entropy (4.71) but higher SWSE (0.65), indicating uncertainty is spread across less meaningful vocabulary. Our model

| Model | Mean Sentence Entropy | SWSE |
|---|---|---|
| RSCaMa | 4.71 | 0.65 |
| RSICCFormer | 0.53 | 0.56 |
| Our Model | 5.58 | 0.60 |

Table 5: Entropy comparison across different models showing SWSE provides more meaningful uncertainty quantification than standard entropy

demonstrates the most effective uncertainty distribution with high standard entropy (5.58) but low SWSE (0.60), suggesting that while the model exhibits overall uncertainty, it maintains confidence in domain-critical terms. This pattern indicates that our image enhancement module, which focuses attention on changed regions, effectively reduces uncertainty for semantically important change-related vocabulary while maintaining appropriate uncertainty for less critical terms.

## 4 CONCLUSION

In this work, we present a comprehensive pipeline that address the critical gap in uncertainity quantification for automated change detection and captioning from remote sensing imagery. We integrate SemantcGraphCD, a novel change detection with dual state space models for efficient spatio-temporal reasoning, complemented by the proposed Semantic Weighted Sentence Entropy (SWSE) for principled uncertainty quantification. Experimental evaluations on the LEVIR-CC and LEVIR-MCI datasets demonstrated that the mode not only achieves state-of-the-art captioning performance but also provides interpretable confidence estimates that address the reliability gap in existing methods. By emphasising domain-relevant vocabulary in uncertainty estimation, SWSE enables more trustworthy decision support in safety-critical applications such as infrastructure monitoring and urban planning. Several interesting directions for future work emerge from the research. First, integration of large language models for more sophisticated temporal reasoning and multi-modal understanding of satellite imagery sequences. Second, extending SWSE to other vision-language tasks in remote sensing where uncertainty quantification is crucial, such as disaster assessment and environmental monitoring. Third, investigating multi-spectral band processing capabilities and developing domain-adaptive semantic weighting schemes for different geographical regions or application domains.

ACKNOWLEDGMENTS

Nil

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
