# OpenReview forum: "Learning to Describe Urban Change: Graph-Guided Detection and spatio-Temporal State Space Model with Uncertainty Estimation"
_ICLR.cc/2026/Conference — ICLR 2026 Conference Withdrawn Submission_

### Official Review · Reviewer_pwYV · 2025-10-21

**Soundness:** 1
**Presentation:** 3
**Contribution:** 2
**Rating:** 4
**Confidence:** 4

**Summary:**

This paper presents a unified pipeline for change detection and captioning from bi-temporal remote sensing images. The key contributions include a novel change detection backbone (SemanticGraphCD) that combines GNNs and CNNs, a State Space Model (SSM)-based captioning module with Spatial Difference-aware SSM (SD-SSM) and Temporal Traversing SSM (TT-SSM) for efficient spatio-temporal modeling, and a new uncertainty metric, Semantic-Weighted Sentence Entropy (SWSE). Evaluated on LEVIR-CC and LEVIR-MCI datasets, the method claims state-of-the-art performance and provides interpretable confidence scores for decision-support systems.

**Strengths:**

The hybrid GNN-SSM architecture and the semantically-weighted uncertainty metric (SWSE) are innovative contributions tailored to the challenges of remote sensing change captioning. The work bridges vision-language modeling and reliability estimation, a crucial step towards trustworthy decision-support systems in safety-critical applications like urban planning and disaster response.

**Weaknesses:**

1.	The paper fails to provide essential implementation details for its core innovations. Specifically: the graph structure and "task-agnostic learning" objective in SemanticGraphCD; the specific parameterization and operational mechanisms of "spatial difference awareness" in SD-SSM and "cross-wise scanning" in TT-SSM. This severely hampers reproducibility and accurate assessment of their novelty.
2.	Computational efficiency claims are not backed by hardware-agnostic metrics (FLOPs, memory) or runtime comparisons.
3.	The ablation study lacks depth, failing to quantify the individual contribution of each proposed module.
4.	Generalization is only tested on two related datasets, lacking cross-dataset evaluation.
5.	The semantic weight values in SWSE appear to be set heuristically without validation (e.g., sensitivity analysis).
6.	There is no mention of code release, which is critical for a work with multiple complex, custom components.

**Questions:**

1.	Could the abstract be revised to be more concise, focusing on the core problem, high-level approach, key results, and impact, while moving detailed module descriptions into the main body?
2.	Implementation of Core Modules: Could you provide:
3.	Detailed architecture diagrams or pseudocode for SemanticGraphCD, specifically explaining the graph construction and the "task-agnostic learning" mechanism?
4.	The precise mathematical formulation or algorithmic description of how "spatial difference awareness" is integrated into the SSM (for SD-SSM) and the exact "cross-scanning" path (for TT-SSM)?
5.	Please report computational metrics (e.g., FLOPs, inference time) comparing your SSM-based captioner with a standard transformer-based baseline.
6.	Could you provide a more comprehensive ablation study that clearly isolates the performance gain from SD-SSM, TT-SSM, and the convolutional focusing module individually?
7.	Have you evaluated the model's performance on other public change detection/captioning datasets to demonstrate broader generalization?
8.	How were the specific semantic weight values in SWSE determined? Please provide an ablation study or theoretical justification for this weighting scheme.
9.	Will the source code and trained models be made publicly available to ensure the reproducibility of these results?

---

### Official Review · Reviewer_m4dD · 2025-10-27

**Soundness:** 2
**Presentation:** 2
**Contribution:** 2
**Rating:** 2
**Confidence:** 4

**Summary:**

In this work, the authors proposes a novel method to generate change detection captions of remote sensing images.
The model takes as input two remote sensing images to produces a change mask. That mask is then used to extract the relevant parts in the original image (Image Enhancement), and passed through a CLIP backbone, State Space Model and finally passed through a Language decoder to generate the change caption.
In addition, the authors introduce "semantic weighted sentence entropy" (SWSE), which ads weights to the words in the generated caption, depending on their importance: (in order of increasing importance) function words, descriptors, natural features, land use, and infrastructure categories respectively.

**Strengths:**

The proposed method seems to be on par or slightly improving over other state of the art methods. The proposed metric of weighting the words in the captions differently is an interesting approach that could potentially lead to more accurate captioning.

**Weaknesses:**

The paper is a little hard to follow, I have a hard time understanding some of the aspects. For example,  the way SemanticGraphCD is introduced makes it seem like it is a well known method, but lacks references. Later, it appears that SemanticGraphCD is new and introduced in the paper. But this is not explicitly stated in the paper.
Overall, the methods are a bit hard to follow, there are a lot of moving parts. Figure 1 helps, but the figure isn't very clear. Each step could be illustrated more and explain what the inputs are, and how they are combined.

I am not sure the paper is actually doing uncertainty quantification. SWSE is introduced as a new metric, which can make sense - although introducing a new metric to evaluate the proposed methods is heavily biased towards the new method. But it doesn't seem like the model is doing uncertainty quantification. Furthermore, how are the weights of SWSE chosen? Again, this could be engineered towards making the model seem more performing than it is.

The proposed method, although sound, has a lot of complicated parts. It is unclear to me how some of these parts help in the performance of the model. As no ablation study has been performed, it is hard to say with certainty that the steps are actually necessary. In addition, SemanticGraphCD seems very complicated just for change detection. How does it compare to simpler methods?

Lastly, the proposed model doesn't actually perform that well when compared to other methods. In table 2, other models perform better in 3 of the six metrics (PSNet performs better than the proposed model for BLEU-2, the bolding is wrong). It doesn't seem like the proposed methods advances the field significantly.

I would also welcome a qualitative assessment with more examples where the model fails.

Minor issues:
- in a lot of places, citations wrongly use the inline style (citet) instead of in parentheses (citep)
- Heading 3.2 is wrong, should read "Quantitative" and not "Qualitative".
- PSNet performs better than the proposed model for BLEU-2, the bolding is wrong
- line 20: missing space "Model(SSM"
- line 245: missing parenthesis around "equation 1"
- Line 385: "The" is not needed at sentence start

**Questions:**

What is the source of the remote sensing images?

---

### Official Review · Reviewer_Tqbn · 2025-11-01

**Soundness:** 1
**Presentation:** 1
**Contribution:** 1
**Rating:** 0
**Confidence:** 4

**Summary:**

This paper proposes a framework for urban change detection and captioning from satellite imagery that integrates a graph-based detection module and a state-space model (SSM)–based captioning system with uncertainty estimation. The method combines: (1) SemanticGraphCD, a graph neural network with adaptive processing between GNN propagation and convolutional operations; (2) Spatial Difference-aware SSM (SD-SSM) and Temporal Traversing SSM (TT-SSM) for enhanced spatio-temporal modeling; and (3) Semantic-Weighted Sentence Entropy (SWSE) for uncertainty quantification, emphasizing domain-relevant vocabulary. The paper claims improved captioning accuracy and interpretable uncertainty estimates compared to previous SSM-based approaches.

**Strengths:**

- The integration of change detection and captioning is a meaningful goal for urban monitoring.

- The idea of incorporating uncertainty quantification through SWSE is potentially useful.

**Weaknesses:**

- The abstract is overly long and fails to clearly state the motivation and core problem, giving an impression of poor focus. The abstract includes too much architectural detail (e.g., descriptions of SemanticGraphCD, SD-SSM, TT-SSM, and SWSE) without first clarifying why these components are needed. It spends nearly the entire length listing modules and mechanisms, but only briefly mentions the underlying problem that current change captioning systems lack uncertainty quantification. As a result, the reader does not immediately understand the main research gap or the practical significance of the work. A clearer abstract should first motivate why uncertainty in change captioning is critical, then summarize the method and findings.

- The citation format is incorrect and unpolished. For example, the paper writes "With the increasing availability of high-resolution satellite data from missions like Landsat Wulder et al. (2019), Sentinel Drusch et al. (2012), and commercial providers Li et al. (2022b) …", where the citations are appended without proper integration into the sentence. This gives the paper an unpolished feel and makes it harder to read.

- Related work on change detection is outdated, missing more recent literature: "Deep learning revolutionized the field
with convolutional neural networks (CNNs) Zhang & Li (2017); Daudt et al. (2018) that could automatically learn hierarchical features, followed by more advanced architectures like U-Net variants Peng et al. (2019) and attention mechanisms Chen & Shi (2020)." There are definitely works about change detection after 2020, such as [1] Chen, Hongruixuan, et al. "ChangeMamba: Remote sensing change detection with spatiotemporal state space model." IEEE Transactions on Geoscience and Remote Sensing 62 (2024): 1-20. [2] Fang, Sheng, Kaiyu Li, and Zhe Li. "Changer: Feature interaction is what you need for change detection." IEEE Transactions on Geoscience and Remote Sensing 61 (2023): 1-11.

- Table 1 lists manually set weights, but no rationale or sensitivity analysis is provided to show how these values affect results.

- Section 3.2's title should be "quantitative results" rather than "qualitative results".

- While the paper lists many baselines (e.g., MCCFormer, PSNet), it does not explain why these particular models were chosen or how they were adapted for the current task.

- No ablation study is provided to isolate contributions of individual modules (SemanticGraphCD, SD-SSM, TT-SSM, SWSE). That would be particularly useful to verify whether each module (SemanticGraphCD, SD-SSM, TT-SSM, SWSE) contributes meaningfully to the overall performance.

- The inclusion of an unfinished Acknowledgements section violates ICLR’s anonymity policy.

- Overall, the writing is hard to follow, and the technical presentation lacks clarity and rigor.

**Questions:**

Please refer to Weaknesses.

While the topic of combining change detection, captioning, and uncertainty estimation is interesting, the paper suffers from serious presentation issues, unclear motivation, lack of experimental rigor, and incomplete justification of design choices. It does not meet ICLR’s standard of clarity or scientific soundness.

---

### Official Review · Reviewer_4f7S · 2025-11-02

**Soundness:** 1
**Presentation:** 1
**Contribution:** 1
**Rating:** 2
**Confidence:** 5

**Summary:**

The manuscript aims to present a comprehensive pipeline for automated change captioning from remote sensing imagery. The pipeline integrates SemanticGraphCD, state space models, Semantic Weighted Sentence Entropy.

**Strengths:**

The paper identifies a critical and underexplored challenge: the absence of uncertainty quantification in existing change captioning systems for remote sensing. This is a well-motivated problem. It is a small contribution to use Semantic Weighted Sentence Entropy for principled uncertainty quantification.

**Weaknesses:**

1、The overall writing and structural logic of the paper are disorganized. The authors present a series of loosely connected motivations and then stack multiple pre-existing components without clear theoretical integration. As a result, the contribution appears engineering-driven rather than scientifically innovative.

2、The paper highlights the SD-SSM and TT-SSM as key innovations for spatial-temporal reasoning. However, these modules have already been introduced in RSCaMa (2024). Simply reusing or reconfiguring such components without conceptual advancement does not constitute a novel contribution.

3、The overall text annotations in the LEVIR-CC and LEVIR-MCI datasets are actually the same. It is recommended to experiment on other datasets, such as the WHU-CDC dataset.

4、While the method section explains why each module (SemanticGraphCD, SD-SSM, TT-SSM, SWSE) is introduced, it lacks details on how they are technically implemented. The paper omits critical aspects such as the mathematical formulation of the SSM components, the adaptive mechanism of SemanticGraphCD, and how SWSE is integrated into the training pipeline. Without these details, the reproducibility and scientific rigor of the method remain questionable.

5、The introduction overlooks several recent advances (2024–2025) in remote sensing change detection and captioning—especially works leveraging state space models (e.g., Mamba variants) and uncertainty-aware captioning frameworks. This omission makes it difficult to assess the originality and relevance of the proposed work within the current research landscape.

6、The paper does not present sufficient ablation studies to quantify the individual contributions of each proposed component (SemanticGraphCD, SD-SSM, TT-SSM, SWSE).

7、The evaluation compares the proposed method mainly with models from 2020–2023 (e.g., PSNet, RSICCFormer), but omits more recent and competitive 2024–2025 methods. Without these, it is difficult to determine whether the method truly achieves state-of-the-art performance.

8、Although Semantic Weighted Sentence Entropy (SWSE) is an interesting idea, it lacks a solid theoretical basis. The weighting scheme appears to be heuristically assigned rather than learned or validated. The paper does not provide empirical evidence showing that SWSE correlates with actual uncertainty or reliability in generated captions.

9、Many of the references in the paper are incorrect, such as the authors and journals listed. Please check carefully.

**Questions:**

1.Can the authors clarify the exact architectural differences between their SD-SSM/TT-SSM and those used in RSCaMa (2024)? If they are reused or modified, please specify the improvements quantitatively or structurally

2.Regarding SWSE, how were the token importance weights determined (e.g., manual heuristics or learned via optimization)?

3.Could the authors provide ablation results isolating the contributions of SemanticGraphCD, SD-SSM, TT-SSM, and SWSE individually? This would clarify which component contributes most to the observed performance gains.

---

### Note · Authors · 2025-11-30

**Comment:**

We would like to withdraw the submission due to a change in our publication plan. After internal discussion, we decided to submit the work to a different venue whose scope and review timeline are better aligned with our current research goals. The withdrawal is not due to any issues with the content of the submission; we simply wish to redirect the manuscript to a conference that is better suited for this line of work.

**Withdrawal Confirmation:**

I have read and agree with the venue's withdrawal policy on behalf of myself and my co-authors.